# Exploring Empirical Rules for Construction Accident Prevention Based on Unsafe Behaviors

Han-Hsiang Wang [1], Jieh-Haur Chen [1,2,3,*], Achmad Muhyidin Arifai [1] and Masoud Gheisari [4]

1   Department of Civil Engineering, National Central University, Jhongli, Taoyuan 320317, Taiwan; hhwang@ncu.edu.tw (H.-H.W.); arifai.ac.id@gmail.com (A.M.A.)
2   Safety and Health Association of Taiwan, Zhunan, Miaoli 350007, Taiwan
3   Research Center of Smart Construction, National Central University, Jhongli, Taoyuan 320317, Taiwan
4   Rinker School of Construction Management, University of Florida, Gainesville, FL 32611, USA; masoud@ufl.edu
*   Correspondence: jhchen@ncu.edu.tw; Tel.: +886-3-422-7151 (ext. 34112); Fax: +886-3-425-0675

**Abstract:** This paper is aimed at exploring rules for construction accident prevention based on unsafe behaviors. The literature review demonstrates a clear connection between construction accident prevention and unsafe behaviors, followed by a 2-year field investigation resulting in 2207 observations based on convenient sampling with 95% confidence and 5% limit of errors in the 50–50 category. There are 80.43% unsafe behaviors categorized into "Regulations for the Occupational Safety and Health Equipment and Measures", where there are 66.37% of regulations and law VII violations, linking fall prevention with the most cases (94.48%) of Fall Protection and Structure Strengthening. The Apriori yields 13 association rules, where the top 3 rules show that 44.11% of the Passage and lighting category is linked to construction equipment inspections; 29.41% of the high-pressure gas category is linked to construction equipment inspections; 100% of the fire prevention category is linked to fire protection unsafe behavior. The findings clarify the association rules that can prevent workers from accidents in construction sites.

**Keywords:** construction safety; association rules; unsafe behavior; data mining

## 1. Introduction

Construction is one of the most dangerous businesses [1,2]. According to research, workers' unsafe behavior is the most significant factor affecting construction safety performance [3–5]. Human errors caused 90% of construction site accidents [6], and 88% of construction engineering mishaps were caused by unsafe human behavior. As a result, safety management in the construction industry is required to prevent unsafe worker behavior [7,8].

The Taiwan construction industry usually has the highest number of deaths regarding occupational safety. For example, there were 316 deaths regarding occupational safety in Taiwan for 2019, where 168 out of 316 deaths (>50%) were related to the Taiwan construction industry. The construction management research community has paid close attention to construction employees' unsafe behavior [9]. Most of the main topics related to unsafe behavior are influencing factors [10,11], formation mechanisms, and pre-control measures of workers' unsafe behaviors. When it comes to pre-control measures, the majority of the studies recommended from the perspective of management as early as possible to restrain the occurrence of unsafe behavior [12]. This result has helped managers increase behavioral safety management levels in the construction sector by identifying critical unsafe behavior as the main control among different workers in different construction phases to prevent and reduce accidents [13]. Studies were conducted on accident cases or near-miss accidents. However, those studies also have limitations on (1) the number of accident cases, (2) connections between safety regulations and unsafe behaviors, and (3) region-wide investigations [5,14,15]. Therefore, the research objectives are to investigate unsafe

behaviors in construction projects and to explore the association rules for construction accident prevention based on unsafe behaviors.

## 2. Literature Review

### 2.1. Safety in Construction

The construction business has the highest injury rate among all industries. Every year, many people suffer from disease or injury and even die due to construction site accidents. For example, there were 316 deaths regarding occupational safety in Taiwan in 2019, where 168 out of 316 deaths (>50%) were related to the Taiwan construction industry. As a result, accident control is critical in the construction sector. Employers must assess risk and take practical actions to safeguard the occupational safety and health of their workers, while limiting risk through constant surveillance and monitoring of areas where accidents are likely to occur [16–21]. Moreover, before starting up or embarking on a construction project, the various construction project management teams are required to prepare a plan for occupational safety, health, and the environment (Health and Safety and Environment Plan), which clarifies the policy of the company in implementing the project in the field of safety and the environment throughout the implementation period until the project is completed and handed over to the owner. By taking actions in reaction to unfavorable injuries or accidents, safety construction has traditionally been measured, recorded, and monitored reactively and documented as cases [16]. Risk analysis, leading indicators, and precursor analysis have all recently been added to the field of construction safety research. Practitioners have turned to expert's assistance, particularly from qualified safety specialists, to attain the goal of being predictive. Even the most experienced safety specialists, however, have limited personal experience with injuries (thousands of working hours) and are subject to a variety of cognitive biases when faced with uncertainty [20,22].

### 2.2. Data Mining in Construction Health and Safety Research

In the construction business, the use of data-mining approaches is becoming more popular. It has attracted the attention of both practitioners and academics around the world [21–24]. It can also be learned from vast amounts of objective, empirical data equating to millions of labor hours [23] to optimize safety management methods [25]. Poh et al. applied a machine learning approach to create a model that can produce a safety leading indicator and be used to forecast building site safety concerns. The industry-recognized Cross Industry Process Model for Data Mining (CRISP-DM) framework was used to systematically apply machine learning (ML) procedures and techniques to a seven-year dataset of a big and reputable construction company. A total of thirty-three input variables (also known as features or independent variables) were identified as part of the research. Thirteen relevant input variables were identified after using the Boruta feature selection technique and a decision tree. Six of the thirteen input variables are project-related (Project Type, Project Ownership, Contract Sum, Completion Percentage, Magnitude of Delay, and Project Manpower) and seven are safety-related (Crane/lifting Operations, Scaffold, Mechanical-Elevated Working Platform, Falling Hazards/Openings, Environmental Management, Good Practices, and Weighted Safety Inspection Score). This emphasizes the significance of good project management in terms of construction safety [21]. Sakhakarmi et al. investigated the effect of feature vectors in classifying likely structural failure scenarios of a complicated scaffold structure using machine learning. Over 23,000 datasets, 23 forms of local and global failures were studied, with 20 original strain characteristics per dataset and 210 enhanced strain features per dataset. Based on the simulation findings, 20 strain features were found to be insufficient to adequately reflect the failure instances caused by the complicated loading combinations given to each of the datasets. The greatest accuracy with the 20 strain feature case was 85.48 percent, whereas it was 96 percent with the 210 strain feature instance. The influence of enhanced strain feature vectors for exactly the same loading scenarios was seen when the prediction accuracy was compared between the two cases [26]. Tixier et al. adopted two cutting-edge ML algorithms: Random Forest

(RF) and Stochastic Gradient Tree Boosting (SGTB). The generated models predict three out of four safety outcomes with high skill, namely injury type, energy type, and body part, using binary fundamental construction qualities as input. This model not only outperforms previous research models in terms of competence but also in terms of the range of outcomes anticipated. It is also worth noting that the SGTB models consistently outperformed their RF counterparts in terms of predictive ability [23].

### 2.3. Association Rule Mining in Construction Health and Safety Research

Association Rule Mining (ARM) is a technique for extracting common patterns, correlations, relationships, and causal structures from datasets stored in a variety of databases and repositories. It has been used to characterize and analyze unknown relationships in data of safety concerns assessment, which is for identifying frequent item sets between uncertain parameters and generating strong association rules from huge datasets, especially auxiliary data from engineering processes. This simple and practical data mining technique can uncover common item sets and association rules across many atypical events that have been tracked [27–30]. The Apriori method was used to analyze the injury data to find statistical correlations between injuries (IBIP). The outcome variable for each IBIP was then employed in logistic regression modeling to find connections between specific road user groups and IBIPs. A total of 48,544 people were studied, with 36,480 (75.1%) having only one injury type and 12,064 (24.9%) having multiple injuries. In the multiple injury sample, data mining revealed 77 IBIPs, 16 of which were linked to only one type of road user. IBIPs and their relationship to the type of road user are one step toward establishing a tool to better comprehend and quantify injury severity and, as a result, improve the evidence base supporting the prioritization of road safety countermeasures [31]. Sivasankaran et al. employed a data mining approach to evaluate vehicle–pedestrian collisions in order to extract information that may be used to improve pedestrian safety. The non-parametric method of association rule mining has the distinct advantage of not limiting the distribution assumption of variables and associations, whereas parametric modeling requires that explanatory variables be independent. Additional association rules uncover significant rules with specific patterns that are not possible in traditional statistical models [32]. Martínez-Rojas et al. examined workplace accidents that occurred on Spanish construction sites while taking into account the workers' nationality. The use of association rules is suggested due to the vast number of incidents and attributes linked with them. Overall, the results show comparable behavior; however, there are some notable variances in the workers' occupations. Furthermore, the findings are consistent with earlier research conducted in other nations. The results of these investigations could be used to develop initiatives to create safer working environments [33]. However, occupational safety accidents still occur frequently on construction sites, both in public and private projects. As a result, construction site workers are reluctant to comply with the laws and regulations, and often make mistakes during construction that do not comply with the laws and regulations, which greatly increases the occurrence rate of accidents [34].

### 3. Site Investigation and Observation Sampling

The study conducted site investigation and observation sampling to investigate the frequency and types of unsafe behaviors that occur at construction sites and to obtain unsafe behavior patterns that occur at the sites through on-site surveys. The target for this study was, based on the concept of convenient sampling [35] to select the largest construction area in Taoyuan City, Taiwan during 2019–2021. There are dozens of construction sites spread out for over a 50,000 square meter area in Taoyuan Technological and Environmental Industrial Park, Taoyuan City, Taiwan. The total construction cost was around 10 billion New Taiwan Dollars (NTD). During 2019–2021, the sites were visited and observed to record unsafe behaviors for on-site workers who conducted their work at job sites, as illustrated in Figure 1. This study adopted the participatory observation method as the main collection method, where the observers served as occupational safety inspectors

for on-site investigation and data collection at the sites, including a major incinerator construction project. During the observation period of over 2 years, the supervisors and occupational safety and health personnel in charge of each area of the site carried out daily inspections to investigate and compile records of errors and violations in accordance with the laws and regulations. These are the Occupational Safety and Health Law, the Occupational Safety and Health Facilities Regulations, the Occupational Safety and Health Facilities Standards, and other related regulations promulgated by the Ministry of Labor Affairs.

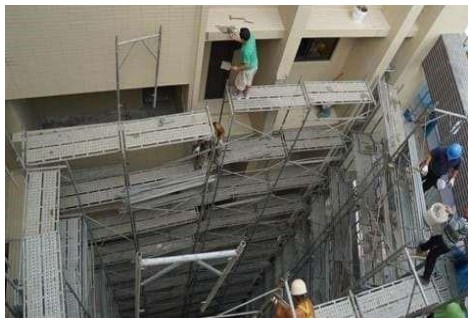 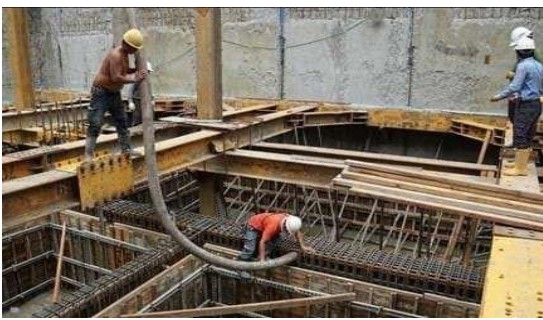

Workers in aloft activities without any protection

Workers in unsafe positions without any protection

**Figure 1.** Demonstrations of unsafe behavior.

Observations were conducted on a floor-by-floor walk-through manner, with outdoor excavation areas and flat areas, indoor areas, such as basements and flat areas, and floor-by-floor walk-throughs. In accordance with the regulations and inspection procedures related to occupational safety and health at construction sites, safety and health personnel were required to conduct toolbox safety education, hazard notification signatures, and toolbox meetings with on-site workers who work in the investigated job sites, not including supervision manpower, and to take photos of occupational safety and health errors seen during construction. Figure 1 demonstrates that workers conducted unsafety behaviors, including no protective actions in aloft activities and grouting. A total of 2207 observations of unsafe behavior were recorded based on convenient sampling with 95% confidence and 5% limits of errors in the 50–50 category [36]. The study then compared and analyzed the contents of the current occupational safety and health laws and regulations with the common unsafe behaviors and explored the discordance between the unsafe behavior and the occupational safety laws and regulations. As a result, unsafe behaviors are classified into categories and violations of the law, as presented in Table 1.

The left column "Unsafe Behaviors" of Table 1 stands for the unsafe behaviors based on 2207 observations collected from the sites. There are 8 major behaviors, including construction equipment inspections, personal misconducts, excavation operations, fall protection and structure strengthening, fire protection, lifting operation, safety management and equipment, and working platform. The middle column "Category" represents the types of unsafe behaviors that can be associated with one or more than one type suggested in the literature [18–21,26,28,30–32]. The right column "Regulations & Laws" are the related regulations and laws associated with occupational safety and health practice in Taiwan. There are 8 major regulations and laws including safety rules for lifting equipment, standards, or labor protection measures for elevated work, labor inspection act, construction safety and health standard, occupational safety and health act, occupational safety and health education and training rules, regulations for the occupational safety and health equipment and measures, and occupational safety and health management measures. These are all considered for association rule mining using Apriori for the next section.

**Table 1.** Unsafe Behavior, Category, and Regulation & Laws.

| | Unsafe Behavior | | Category | | Regulations & Laws |
|---|---|---|---|---|---|
| 1 | Construction equipment inspections | A | Construction method | I | Safety rules for lifting equipment |
| 2 | Personal misconducts | B | License | II | Standards for labor protection measures for elevated work |
| 3 | Excavation operations | C | Hazard | III | Labor inspection act |
| 4 | Fall protection and structure strengthening | D | Self-inspection | IV | Construction safety and health standard |
| 5 | Fire protection | E | Confined space operation | V | Occupational safety and health act |
| 6 | Lifting operation | F | Passage and lighting | VI | Occupational safety and health education and training rules |
| 7 | Safety management and equipment | G | Fire prevention | VII | Regulations for the occupational safety and health equipment and measures |
| 8 | Working platform | H | Fall prevention | VIII | Occupational safety and health management measures |
| | | I | Protective facilities | | |
| | | J | Scaffolding | | |
| | | K | Personal protective gears | | |
| | | L | Lifting and hanging work | | |
| | | M | High-pressure gas | | |
| | | N | Educational training | | |
| | | O | Alcoholic beverages | | |
| | | P | Excavation operations | | |
| | | Q | Machinery | | |

## 4. Association Rule Mining

ARM's objective is to achieve strong rules in databases using some interesting measurements. ARM requires constraints on multiple measures of significance to be identified to design and select rules from a set of options. Two key characteristics were used to assess the strength of an association rule: support and confidence. The frequency with which a specific rule appeared in the database being mined was referred to as support. The number of times of a given rule turned out to be true in practice, which is referred to as confidence. A rule may appear to have a strong association in data collection because it appeared frequently but might emerge much less frequently when applied. This would be a situation where there were numerous supports, but not a lot of confidence. In contrast, a rule might not stand out in data collection, but further research revealed that it frequently occurred. This is a situation where there is a lot of confidence but not a lot of support. These metrics aid analysts in distinguishing causation from correlation and determining the usefulness of a rule. The lift value, or confidence-to-support ratio, is a third value component. There is a negative correlation between datasets if the lift value is negative. There is a positive correlation if the value is positive, and there is no correlation if the ratio is equal to 1. With the Apriori setting for confidence greater than 90%, Figure 2 demonstrates that unsafe behaviors are structured based on regulation and laws.

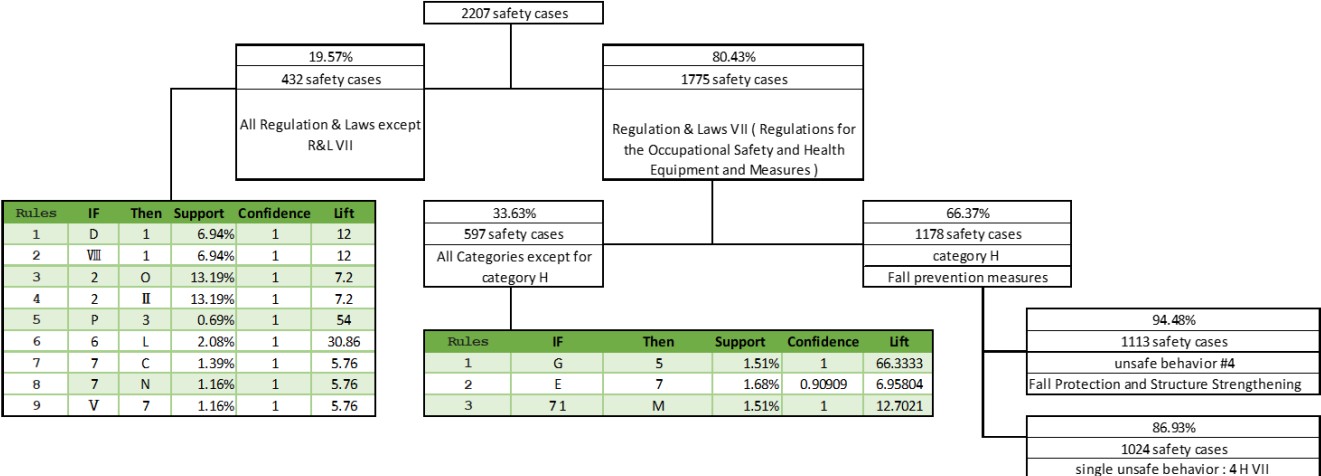

**Figure 2.** Structured unsafe behavior based on regulation and laws.

## 5. Research Findings and Discussion

The occurrence of unsafe behavior is often caused by a number of factors, including the manufacturer's own costs, the convenience of the manufacturer, the difficulty of construction due to regulations, or the convenience of the laborers themselves, all of which can cause errors. The statistical categories are shown in Figure 3.

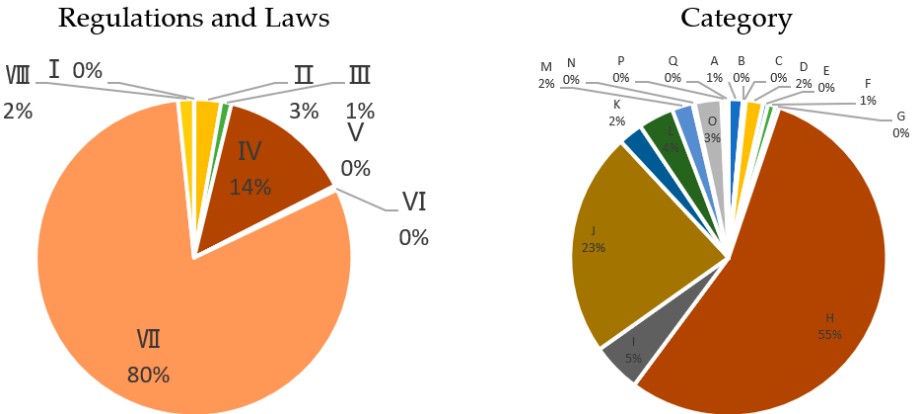

**Figure 3.** Regulations and laws, and category.

Figure 3 shows the categories of each unsafe behavior that were sorted into construction method and self-inspection. The statistics show that the first two most frequent unsafe behavior samples in construction sites are fall prevention (55%) and Scaffolding (23%). Figure 3 also indicates that 80% of unsafe behaviors are due to the violation of regulation & laws VII, which is the Regulation for the Occupational safety and Health Equipment and Measures. More specifically, reinforcing steel and formwork workers are the most common types of unsafe behavior violating this regulation. Based on this finding, unsafe behaviors are structured based on regulation and laws, as shown in Figure 2, where the Apriori setting for confidence is greater than 90%.

There were 1775 unsafe behaviors (or 80.43% of 2207 reports) categorized into "Regulations for the Occupational Safety and Health Equipment and Measures". Furthermore, in this category, there were 1178 unsafe behaviors (66.37% of regulation and laws VII violation) that were linked to category H with the most cases of Fall Protection and Structure Strengthening (94.48%). This result is the most reliable because it comes from the majority of unsafe behavior data. The rest of the data in this category (33.63%) are linked to other categories, except for category H. There are some rules as a result of the Apriori algorithm. Rule 1 shows that 44.11% of the Passage and lighting category is linked to construction equipment

inspections. Rule 2 also shows that 29.41% of High-pressure gas is linked to construction equipment inspections. Rule 3 indicates that 100% of the Fire prevention category is linked to Fire Protection unsafe behavior, while in rule 4 it can be seen that 71.42% of Lifting and hanging work is linked to Lifting Operation. Rule 5 is also linked to Lifting Operation unsafe behavior, which has a 23.8% confidence value of the Machinery category.

For unsafe behavior that does not belong to regulation and laws VII, the result indicates that only regulation and laws II and VIII have significant support and confidence values. Rule 1 shows that 100% of Self-inspection affects construction equipment inspections, which refers to any error in the Self-inspection would be linked to Construction equipment inspections. This rule is also linked to rule 2, which states that 100% of the Occupational Safety and Health Management Measures violation is about Construction equipment inspections. In Rule 3, it can be seen that 100% of Personal misconduct is about alcoholic beverages. Personal misconduct is also 100% linked to Standards for Labor Protection Measures for Elevated Work violation (Rule 4). It can be concluded that any unsafe behavior in a construction project would be attributed to alcoholic beverages and the violation of the regulation and laws II; Standards for Labor Protection Measures for Elevated Work. Rule 5 and Rule 6 have a relatively low number of support (below 5%), but they have the perfect score of confidence (100%). Rule 5 indicates that an excavation operations category originates from excavation operations unsafe behavior, while Rule 6 indicates that a Lifting Operation unsafe behavior would affect the Lifting and hanging work category. As stated above, based on statistical and Apriori algorithm output, significant unsafe behavior patterns are as follows:

1.  Regulations for the Occupational Safety and Health Equipment and Measures → Fall prevention → Fall Protection and Structure Strengthening
2.  Fire prevention → Fire Protection.
3.  Confined space operation → Safety management and equipment.
4.  Safety management and equipment, and Construction equipment inspections → High-pressure gas.
5.  Self-inspection → Construction equipment inspections.
6.  Occupational Safety and Health Management Measures → Construction equipment inspections.
7.  Personal misconduct → Alcoholic beverages.
8.  Personal misconduct → Standards for Labor Protection Measures for Elevated Work.
9.  Excavation operation → Excavation operation.
10. Lifting operation → Lifting and hanging work
11. Safety management and equipment → Hazard.
12. Safety management and equipment → Educational training.
13. Occupational Safety and Health Act → Safety management and equipment.

## 6. Conclusions

The case study of construction sites regarding accident prevention in Taiwan aims to investigate the pattern of unsafe behaviors to uncover the relationship between unsafe behaviors and the violation of regulations and laws. A total of 2207 observations of unsafe behavior were recorded based on convenient sampling with 95% confidence and 5% limit of errors in the 50–50 category. There were 1775 unsafe behaviors (or 80.43% out of 2207 reports) categorized into "Regulations for the Occupational Safety and Health Equipment and Measures". Furthermore, in this category, there were 1178 unsafe behaviors (66.37% of regulation and laws VII violation) that were linked to category H with the most cases of Fall Protection and Structure Strengthening (94.48%). These are the most reliable because they come from the majority of unsafe behavior data. The Apriori yielded 13 association rules. For example, Rule 1 shows that 44.11% of Passage and lighting category is linked to Construction equipment inspections. Rule 2 also shows that 29.41% of High-pressure gas is linked to Construction equipment inspections. Rule 3 indicates that 100% of the Fire prevention category is linked to Fire Protection unsafe behavior, while in

rule 4 it can be seen that 71.42% of Lifting and hanging work is linked to Lifting Operation. Rule 5 is also linked to Lifting Operation unsafe behavior, which has a 23.8% confidence value of the Machinery category. The contribution lies in the association rules that guide management practices for construction job sites.

The current domestic occupational safety laws and regulations were designed to protect the lives of workers; however, the regulations themselves affect construction efficiency and increase construction costs, resulting in domestic workers in construction sites being unable to properly follow SOPs. Findings from this study provide priority and suggest that those 13 rules should serve as guidelines to set up priorities in order to simplify the current SOPs. It especially takes place when the safety budget for construction projects is limited, implying that safety management has priority in practice. Future work can focus on how to classify safety inspections to practice more efficiently. Creating a system that conducts automation detection applied to workers' behaviors to send out warnings to those behaving unsafe postures or patterns is also feasible and contributable to the industry.

**Author Contributions:** Conceptualization J.-H.C.; Methodology, J.-H.C. and A.M.A.; Validation, M.G. and J.-H.C.; Resources, J.-H.C. and H.-H.W.; Writing—original draft preparation, A.M.A. and J.-H.C.; writing—review and editing, J.-H.C. and H.-H.W. All authors have read and agreed to the published version of the manuscript.

**Funding:** This paper was partly supported by the Ministry of Science and Technology (MOST), Taiwan, for promoting the academic excellence of universities under grant numbers [MOST 110-2221-E-008-052-MY3]; [MOST 110-2622-E-008-018-CC2]; [MOST 110-2221-E-008 -036 -MY2]; [MOST 109-2622-E-008-018-CC2]; and [MOST 108-2221-E-008-002-MY3].

**Institutional Review Board Statement:** Not applicable.

**Informed Consent Statement:** Not applicable.

**Data Availability Statement:** The data presented in this study are available on request from the corresponding author. The data are not publicly available due to the sponsored investigation.

**Conflicts of Interest:** The authors declare no conflict of interest.

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
