# Peer review of "Exploring Empirical Rules for Construction Accident Prevention Based on Unsafe Behaviors"

_sustainability, doi:10.3390/su14074058_

Round 1

Reviewer 1 Report

The study explores rules for construction accident prevention based on unsafe behaviors in Taiwan construction industry, where 2207 observations were collected practically on site and analyzed using data mining technique. Despite the hard work paid to collect the dataset, major comments were found in the paper, and should be addressed before the final decision.

  1. The abstract should be modified to have some critical results related to the rules that were analyzed.
  2. In the introduction, the review part should have the limitations and advantages of the other studies in accordance with the behavioral factors. The authors did not clarify this point. For example, the last three lines of the 2.3 section should be supported by some literatures.
  3. For the inspection process, the framework of the inspection procedures should be shown through figures or tables.
  4. In section 3.1, the authors mentioned that some photos were taken of the errors occurred, the best case is to provide the paper with some of these photos or schemes.
  5. Section 4 figures are low quality. Fig1 and Fig2.
  6. In section 4.1, the paper lacks the frequency tables of the detected accidents. So, the authors are required to add statistical tables of the case studies. For example, in the conclusion the sentence “ More specifically, reinforcing steel and formwork workers are the most common types of unsafe behavior violating this regulation” was not supported by the data during the paper discussion.
  7. Please clarify all the abbreviations in the paper.
  8. Some studies are required to be added to enrich the literature.
  • Causal factors and risk assessment of fall accidents in the U.S. construction industry: A comprehensive data analysis (2000–2020). https://doi.org/10.1016/j.ssci.2021.105537
  • Tag and IoT based safety hook monitoring for prevention of falls from height. https://doi.org/10.1016/j.autcon.2022.104153

Author Response

Reviewer: 1

The study explores rules for construction accident prevention based on unsafe behaviors in Taiwan construction industry, where 2207 observations were collected practically on site and analyzed using data mining technique. Despite the hard work paid to collect the dataset, major comments were found in the paper, and should be addressed before the final decision.

  1. The abstract should be modified to have some critical results related to the rules that were analyzed.

Responses: We have revised and reorganized the abstract by “The paper is aimed at exploring rules for construction accident prevention based on unsafe behaviors. Literature review demonstrates clearly connection between construction accident prevention and unsafe behaviors, followed by a 2-year field investigation resulting in 2,207 observations based on convenient sampling with 95% confidence and 5% limits of errors in 50-50 category. There are 80.43% unsafe behaviors categorized into “Regulations for the Occupational Safety and Health Equipment and Measures” where there are 66.37% of regulation and laws VII violation, linking to fall prevention with the most cases (94.48%) of Fall Protection and Structure Strengthening. The Apriori yields 13 association rules where top 3 rules show that 44.11% of Passage and lighting category is linked to construction equipment inspections; 29.41% of high-pressure gas category is linked to construction equipment inspections; 100% of Fire prevention category is linked to Fire Protection unsafe behavior. The findings clarify the association rules that can prevent workers from accidents in construction sites.”. Please see the highlighted parts on Page 1.

  1. In the introduction, the review part should have the limitations and advantages of the other studies in accordance with the behavioral factors. The authors did not clarify this point. For example, the last three lines of the 2.3 section should be supported by some literatures.

Responses: We have clarified the limitations from other studies as the paper’s motivation by organizing the sentences to “This result has helped managers increase behavioral safety management levels in the construction sector by identifying critical unsafe behavior as a main control among different workers in different construction phases to prevent and reduce accidents [13]. Studies were conducted in accident cases or near-miss accidents. However, those studies also have limitations on: (1) the amount of accident cases, (2) connections between safety regulations and unsafe behaviors, and (3) region-wide investigation [5, 14, 15].” Please see the highlighted parts on Pages 1 and 2.

We have added the citation for the sentence by “As a result, construction site workers are reluctant to comply with the laws and regulations, and often make mistakes during construction that do not comply with the laws and regulations, which greatly increases the occurrence rate of accidents [31].”. Please see the highlighted parts on Page 3.

  1. For the inspection process, the framework of the inspection procedures should be shown through figures or tables.

Responses: We have named the Section 3 to “Site investigation and observation sampling”. In order to clarify  the framework of the inspection procedures we have rewritten the first paragraph by “The study conducted the site investigation and observation sampling to investigate the frequency and types of unsafe behaviors that occur at the construction sites, and to obtain the unsafe behavior patterns that occur at the sites through on-site surveys. The target for this study was, based on the concept of the convenient sampling [32], to select the largest construction area in Taoyuan City, Taiwan during 2019-2021. There are dozens of construction sites spreading out for over 50,000 meter square area in Taoyuan Technological and Environmental Industrial Park, Taoyuan City, Taiwan. During 2019-2021, the sites were visited and observed to record unsafe behaviors for frontline workers. Figure 1 demonstrates that workers conducted unsafety behaviors including no protective actions in aloft activities and grouting. This study adopted the participatory observation method as the main collection method, and conducted on-site investigation and data collection at the sites including a major incinerator construction project. During the observation period of over 2 years, the supervisors and occupational safety and health personnel in charge of each area of the site carried out daily inspections to investigate and compiled records of errors and violations in accordance with the Occupational Safety and Health Law, the Occupational Safety and Health Facilities Regulations, the Occupational Safety and Health Facilities Standards, and other related regulations promulgated by the Ministry of Labor Affairs”. Figure 1 has been added to demonstrate unsafe behaviors.  Please see the highlighted parts and Figure 1 on Pages 3 and 4.

  1. In section 3.1, the authors mentioned that some photos were taken of the errors occurred, the best case is to provide the paper with some of these photos or schemes.

Responses: We have renamed the Section 3 to “Site investigation and observation sampling”. In order to clarify  the framework of the inspection procedures, we have rewritten the first paragraph by “The study conducted the site investigation and observation sampling to investigate the frequency and types of unsafe behaviors that occur at the construction sites, and to obtain the unsafe behavior patterns that occur at the sites through on-site surveys. The target for this study was, based on the concept of the convenient sampling [32], to select the largest construction area in Taoyuan City, Taiwan during 2019-2021. There are dozens of construction sites spreading out for over 50,000 meter square area in Taoyuan Technological and Environmental Industrial Park, Taoyuan City, Taiwan. During 2019-2021, the sites were visited and observed to record unsafe behaviors for frontline workers. Figure 1 demonstrates that workers conducted unsafety behaviors including no protective actions in aloft activities and grouting. This study adopted the participatory observation method as the main collection method, and conducted on-site investigation and data collection at the sites including a major incinerator construction project. During the observation period of over 2 years, the supervisors and occupational safety and health personnel in charge of each area of the site carried out daily inspections to investigate and compiled records of errors and violations in accordance with the Occupational Safety and Health Law, the Occupational Safety and Health Facilities Regulations, the Occupational Safety and Health Facilities Standards, and other related regulations promulgated by the Ministry of Labor Affairs”. Figure 1 has been added to demonstrate unsafe behaviors.  Please see the highlighted parts and Figure 1 on Pages 3 and 4.

  1. Section 4 figures are low quality. Fig1 and Fig2.

Responses: We have redrawn all figure to increase the resolution. Please see Figures 2 and 3 on Page 6.

  1. In section 4.1, the paper lacks the frequency tables of the detected accidents. So, the authors are required to add statistical tables of the case studies. For example, in the conclusion the sentence “ More specifically, reinforcing steel and formwork workers are the most common types of unsafe behavior violating this regulation” was not supported by the data during the paper discussion.

Responses: We have added descriptions for Figure 2 explaining how it came from by “Figure 2 also indicates that 80% of unsafe behaviors are due to the violation of regulation & laws VII, which is Regulation for the Occupational safety and Health Equipment and Measures. More specifically, reinforcing steel and formwork workers are the most com-mon types of unsafe behavior violating this regulation.”. Please see the highlighted parts on Page 6.

  1. Please clarify all the abbreviations in the paper.

Responses: We have clarified all the abbreviations by adding their full names: Cross Industry Process Model for Data Mining (CRISP-DM) and Machine Learning (ML). Please see the highlighted parts on Page 2.

  1. Some studies are required to be added to enrich the literature.

Causal factors and risk assessment of fall accidents in the U.S. construction industry: A comprehensive data analysis (2000–2020). https://doi.org/10.1016/j.ssci.2021.105537

Tag and IoT based safety hook monitoring for prevention of falls from height. https://doi.org/10.1016/j.autcon.2022.104153

Responses: We have added these 2 references in the literature review. Please see the highlighted parts on Page 2.

Reviewer 2 Report

Brief summary

The paper tries to explore rules for construction accident prevention based on unsafe behaviours in Taiwan construction industry.

Strength of paper

The strength of the paper lies in the fact that its communication is good. The language is simple and easy to follow.

Specific comments in the form of weaknesses

Unfortunately, there are serious flaws, especially in the methodology and results and discussion which must be well addressed. The following comments could help shape up the paper.

  1. The introduction is concise; however, no clear problem that has merited this study has been stated and justified.
  2. Will the findings apply to the construction industry worldwide? Taking into consideration the unique nature of the industry in a specific country, it will be good if the problem is contextualized in this case to the Taiwan environment.
  3. Clear gaps must be stated and a clear research question posed as well.
  4. Under Section 2.1, lines 2-3 must be well referenced. If possible, provide some statistics on the number of accidents and deaths in the industry yearly.
  5. The literature review could also be carved to include health and safety issues in Taiwan construction industry.
  6. It will be good if the Section 2.2 is retitled as ‘Data mining in construction health and safety research’. The contents must then be skewed towards the topic. In its current form the topic deviates from the theme of the paper. The same comment applies to Section 2.3.
  7. The research methodology, i.e., Section 3 must be well expanded. There is the need to state and justify the research approach and strategies adopted for this study.
  8. The population of the study must be well defined. The population size must be stated.
  9. Why were huge areas of construction sites in Taoyuan City chosen over similar cities in Taiwan?
  10. There is the need to clearly and scientifically determine the sample size of the study. The sampling technique used to select the samples must then be well described.
  11. Who were the frontline workers that were observed?
  12. There is the need to introduce readers to what is meant by the Participatory observation methods.
  13. How many construction sites are under investigation? How many observations were carried out on each of the sites and why?
  14. Table 1 must be well discussed in the text. Where were these unsafe behaviors obtained from? Were they through a literature review? If so the sources of such information must be provided.
  15. Section 3.2 is not well integrated in the methodology. It seems to be a stand-alone.
  16. I would suggest to the authors to make the methodology extra robust. The following sub-sections could be covered under the methodology: 3.1 Research approach and strategy (state the research approach and strategy used and justify why they were selected over others); 3.2 study design; and 3.3 Data analysis (Describe how your data was analyzed). On the other hand, if information was obtained by observation, an appropriate sub-section must be created for it and it must be well discussed.
  17. The results are not well presented. The results currently presented do not bring out any meaning and it fails to achieve the specific objectives. The section 4 must be well checked. Again, the results must be appropriately discussed. If possible present and discuss them under the two specific objectives.
  18. Th conclusion fails to tie together the various elements of the paper. There is the need to discuss the theoretical and practical implications of the paper. Also, key limitations must be stated and recommendations for further studies must be made.
  19. The paper with the details below could help improve the literature review and other sections as well:

Adinyira, E., Manu, P., Agyekum, K., Mahamadu, A. M., & Olomolaiye, P. (2020). Violent Behaviour on Construction Sites: Structural Equation Modelling of its Impact on Unsafe Behaviour Using Partial Least Squares. Engineering Construction and Architectural Management. https://doi.org/10.1108/ECAM-09-2019-0489

Author Response

Reviewer: 2

Unfortunately, there are serious flaws, especially in the methodology and results and discussion which must be well addressed. The following comments could help shape up the paper.

  1. The introduction is concise; however, no clear problem that has merited this study has been stated and justified.

Responses: We have revised the problem statement by “When it comes to pre-control measures the majority of the studies recommended from perspectives of management to as early as possible to restrain the occurrence of unsafe behavior [12]. This result has helped managers increase behavioral safety management levels in the construction sector by identifying critical unsafe behavior as a main control among different workers in different construction phases to prevent and reduce accidents [13]. Studies were conducted in accident cases or near-miss accidents. However, those studies also have limitations on: (1) the amount of accident cases, (2) connections between safety regulations and unsafe behaviors, and (3) region-wide investigation [5, 14, 15].” Please see the highlighted parts on Page 1.

  1. Will the findings apply to the construction industry worldwide? Taking into consideration the unique nature of the industry in a specific country, it will be good if the problem is contextualized in this case to the Taiwan environment.

Responses: We have added more discussions for the Taiwan environment by “There were 1,775 unsafe behaviors (or 80.43% out of 2,207 reports) categorized into “Regulations for the Occupational Safety and Health Equipment and Measures”. Furthermore, in this category there were 1,178 unsafe behaviors (66.37% of regulation and laws VII violation) that were linked to category H with the most cases of Fall Protection and Structure Strengthening (94.48%). This result is the most reliable because it comes from the majority of the unsafe behavior data. The rest of the data in this category (33.63 %) are linked to other categories except for category H. There are some rules as the result from the Apriori algorithm. Rule 1 shows that 44.11% of Passage and lighting category is linked to construction equipment inspections. Rule 2 also shows that 29.41% of High-pressure gas is linked to construction equipment inspections. Rule 3 indicates that 100% of Fire prevention category is linked to Fire Protection unsafe behavior, while in rule 4 it can be seen that 71.42% of Lifting and hanging work is linked to Lifting Operation. Rule 5 is also linked to Lifting Operation unsafe behavior which has 23.8% as confidence value of Machinery category.

For the unsafe behavior that does not belong to regulation and laws VII, the result indicates that only regulation and laws II and VIII have significant support and confidence values. Rule 1 shows that 100% of Self-inspection affects the construction equipment inspections, which refers to any error in the Self-inspection would be linked to Construction equipment inspections. This rule is also linked to rule 2, that 100% of Occupational Safety and Health Management Measures violation is about Construction equipment inspections. In Rule 3, it can be seen that 100% of Personal misconduct is about alcoholic beverages. The Personal misconduct is also 100% linked to Standards for Labor Protection Measures for Elevated Work violation (Rule 4). This can be concluded that any unsafe behavior in a construction project would attribute to alcoholic beverages and the violation of the regulation and laws II; Standards for Labor Protection Measures for Elevated Work. Rule 5 and Rule 6 have relatively low number of support (below 5%), but it has the perfect score of confidence (100%). Rule 5 indicates that an excavation operations category originates from excavation operations unsafe behavior; while Rule 6 indicates that a Lifting Operation unsafe behavior would affect Lifting and hanging work category. As stated above, based on statistical and Apriori algorithm output, significant unsafe behavior patterns are as follows:

Regulations for the Occupational Safety and Health Equipment and Measures ® fall prevention measures ® Fall Protection and Structure Strengthening

  1. Regulations for the Occupational Safety and Health Equipment and Measures ® Fall prevention ® Fall Protection and Structure Strengthening
  2. Fire prevention ® Fire Protection.
  3. Confined space operation ® Safety management and equipment.
  4. Safety management and equipment, and Construction equipment inspections ®High-pressure gas.
  5. Self-inspection ® Construction equipment inspections.
  6. Occupational Safety and Health Management Measures ® Construction equipment inspections.
  7. Personal misconduct ® Alcoholic beverages.
  8. Personal misconduct ® Standards for Labor Protection Measures for Elevated Work.
  9. Excavation operation ® Excavation operation.
  10. Lifting operation ®Lifting and hanging work
  11. Safety management and equipment ®
  12. Safety management and equipment ® Educational training.
  13. Occupational Safety and Health Act ® Safety management and equipment.

Please see the highlighted parts on Pages 6 and 7.

  1. Clear gaps must be stated and a clear research question posed as well.

Responses: We have revised the problem statement by “When it comes to pre-control measures the majority of the studies recommended from perspectives of management to as early as possible to restrain the occurrence of unsafe behavior [12]. This result has helped managers increase behavioral safety management levels in the construction sector by identifying critical unsafe behavior as a main control among different workers in different construction phases to prevent and reduce accidents [13]. Studies were conducted in accident cases or near-miss accidents. However, those studies also have limitations on: (1) the amount of accident cases, (2) connections between safety regulations and unsafe behaviors, and (3) region-wide investigation [5, 14, 15].” Please see the highlighted parts on Page 1.

  1. Under Section 2.1, lines 2-3 must be well referenced. If possible, provide some statistics on the number of accidents and deaths in the industry yearly.

Responses: We have added the yearly death number in Taiwan for 2019 by “For example, there were 316 deaths regarding occupational safety in Taiwan for 2019 where 168 out of 316 deaths (> 50%) were related to the Taiwan construction industry.” Please see the highlighted parts on Page 2.

  1. The literature review could also be carved to include health and safety issues in Taiwan construction industry.

Responses: We have briefly added the health and safety issues in Taiwan construction industry in the introduction section to emphasize its importance by “The Taiwan construction industry usually has the highest death number regarding occupational safety. For example, there were 316 deaths regarding occupational safety in Taiwan for 2019 where 168 out of 316 deaths (> 50%) were related to the Taiwan construction industry. The construction management research community has paid close attention to construction employees' unsafe behavior [9].”. Please see the highlighted parts on Page 1.

  1. It will be good if the Section 2.2 is retitled as ‘Data mining in construction health and safety research’. The contents must then be skewed towards the topic. In its current form the topic deviates from the theme of the paper. The same comment applies to Section 2.3.

Responses: We have renamed Sections 2.2 and 2.3 to “Data mining in construction health and safety research” and “Association rule mining in construction health and safety research”, respectively. Please see the highlighted parts on Page 2.

  1. The research methodology, i.e., Section 3 must be well expanded. There is the need to state and justify the research approach and strategies adopted for this study.

Responses: We have expended Section 3 Methodology to Section 3 Site Investigation and Observation Sampling and Section 4 Association Rule Mining. We also have re-organized the entire contents. Please see the highlighted parts on Pages 3-6.

  1. The population of the study must be well defined. The population size must be stated.

Responses: We have explained the population for data sampling by “The study conducted the site investigation and observation sampling to investigate the frequency and types of unsafe behaviors that occur at the construction sites, and to obtain the unsafe behavior patterns that occur at the sites through on-site surveys. The target for this study was, based on the concept of the convenient sampling [34], to select the largest construction area in Taoyuan City, Taiwan during 2019-2021. There are dozens of construction sites spreading out for over 50,000 meter square area in Taoyuan Technological and Environmental Industrial Park, Taoyuan City, Taiwan. The total construction cost was around 10 billion New Taiwan Dollars (NTD). During 2019-2021, the sites were visited and observed to record unsafe behaviors for frontline workers. This study adopted the participatory observation method as the main collection method, and conducted on-site investigation and data collection at the sites including a major incinerator construction project. During the observation period of over 2 years, the supervisors and occupational safety and health personnel in charge of each area of the site carried out daily inspections to investigate and compiled records of errors and violations in accordance with the laws and regulations.” Please see the highlighted parts on Page 4.

  1. Why were huge areas of construction sites in Taoyuan City chosen over similar cities in Taiwan?

Responses: The huge areas of construction sites in Taoyuan City were chosen due to the convenient data sampling and relatively large at that time. We have added explanations by “The study conducted the site investigation and observation sampling to investigate the frequency and types of unsafe behaviors that occur at the construction sites, and to obtain the unsafe behavior patterns that occur at the sites through on-site surveys. The target for this study was, based on the concept of the convenient sampling [34], to select the largest construction area in Taoyuan City, Taiwan during 2019-2021. There are dozens of construction sites spreading out for over 50,000 meter square area in Taoyuan Technological and Environmental Industrial Park, Taoyuan City, Taiwan. The total construction cost was around 10 billion New Taiwan Dollars (NTD).” Please see the highlighted parts on Page 4.

  1. There is the need to clearly and scientifically determine the sample size of the study. The sampling technique used to select the samples must then be well described.

Responses: We have added the citations that adopted data sampling technique based on convenient sampling by “A total of 2,207 observations of unsafe behavior were recorded based on convenient sampling with 95% confidence and 5% limits of errors in 50-50 category [35, 36].”. Please see the highlighted parts on Page 4.

  1. Who were the frontline workers that were observed?

Responses: We have renamed frontline workers to on-site workers who work in the investigated job sites not including supervision manpower. Please see the highlighted parts on Page 4.

  1. There is the need to introduce readers to what is meant by the Participatory observation methods.

Responses: We have added explanations for participatory observation methods by “This study adopted the participatory observation method as the main collection method where the observers served as occupational safety inspectors conducted on-site investigation and data collection at the sites including a major incinerator construction project.” Please see the highlighted parts on Page 4.

  1. How many construction sites are under investigation? How many observations were carried out on each of the sites and why?

Responses: We have added more descriptions for the investigated sites and how to conduct the observations by “The study conducted the site investigation and observation sampling to investigate the frequency and types of unsafe behaviors that occur at the construction sites, and to obtain the unsafe behavior patterns that occur at the sites through on-site surveys. The target for this study was, based on the concept of the convenient sampling [34], to select the largest construction area in Taoyuan City, Taiwan during 2019-2021. There are dozens of construction sites spreading out for over 50,000 square meters of area in Taoyuan Technological and Environmental Industrial Park, Taoyuan City, Taiwan. The total construction cost was around 10 billion New Taiwan Dollars (NTD). During 2019-2021, the sites were visited and observed to record unsafe behaviors for on-site workers, who conducted their work at job sites illustrated in Figure 1.” Please see the highlighted parts on Page 4.

  1. Table 1 must be well discussed in the text. Where were these unsafe behaviors obtained from? Were they through a literature review? If so the sources of such information must be provided.

Responses: We have added descriptions for Table 1 by “The left column “Unsafe Behaviors” of Table 1 stands for the unsafe behaviors based on 2,207 observations collected from the sites. There are 8 major behaviors including construction equipment inspections, personal misconducts, excavation operations, fall protection and structure strengthening, fire protection, lifting operation, safety management and equipment, and working platform. The middle column “Category” represents the types for unsafe behaviors which can be associated with one or more than one types suggested from the literature [18-21, 26, 28, 30-32]. The right column “Regulations & Laws” are the related regulations and laws associated with occupational safety and health practice in Taiwan. There are 8 major regulations and laws including safety rules for lifting equipment, standards or labor protection measures for elevated work, labor inspection act, construction safety and health standard, occupational safety and health act, occupational safety and health education and training rules, regulations for the occupational safety and health equipment and measures, and occupational safety and health management measures. These are all considered for the association rule mining using Apriori for the next section.” Please see the highlighted parts on Page 5.

  1. Section 3.2 is not well integrated in the methodology. It seems to be a stand-alone.

Responses: We have renamed the section to 4. Association Rule Mining. More discussions regarding association rules have been added in the next section. Please see the highlighted parts on Pages 5-7.

  1. I would suggest to the authors to make the methodology extra robust. The following sub-sections could be covered under the methodology: 3.1 Research approach and strategy (state the research approach and strategy used and justify why they were selected over others); 3.2 study design; and 3.3 Data analysis (Describe how your data was analyzed). On the other hand, if information was obtained by observation, an appropriate sub-section must be created for it and it must be well discussed.

Responses: We have reorganized and renamed the sections by “1. Introduction, 2. Literature Review (2.1. Safety in construction, 2.2. Data mining in construction health and safety research, 2.3. Association rule mining in construction health and safety research), 3. Site Investigation and Observation Sampling, 4. Association Rule Mining, 5. Research Findings and Discussions, 6. Conclusions”. Please see the highlighted parts on Pages 3-8.

  1. The results are not well presented. The results currently presented do not bring out any meaning and it fails to achieve the specific objectives. The section 4 must be well checked. Again, the results must be appropriately discussed. If possible present and discuss them under the two specific objectives.

Responses: We have renamed Section 4 to 5. Research Findings and Discussions and reorganized the discussions by “The occurrence of unsafe behavior is often caused by a number of factors, including the manufacturer's own costs, the convenience of the manufacturer, the difficulty of construction due to regulations, or the convenience of the laborers themselves, all of which can cause errors. The statistical categories are shown in Figure 3. Figure 3 shows the categories of each unsafe behavior that were sorted into construction method, self-inspection, and (one more example), etc. The statistics show that the first two most frequent unsafe behavior samples in construction sites are fall prevention (55%) and Scaffolding (23%). Figure 3 also indicates that 80% of unsafe behaviors are due to the violation of regulation & laws VII, which is Regulation for the Occupational safety and Health Equipment and Measures. More specifically, reinforcing steel and formwork workers are the most common types of unsafe behavior violating this regulation. Based on this finding, the unsafe behaviors are structured based on regulation and laws as shown in Figure 2 where the Apriori setting for confidence is greater than 90%.

There were 1,775 unsafe behaviors (or 80.43% out of 2,207 reports) categorized into “Regulations for the Occupational Safety and Health Equipment and Measures”. Furthermore, in this category there were 1,178 unsafe behaviors (66.37% of regulation and laws VII violation) that were linked to category H with the most cases of Fall Protection and Structure Strengthening (94.48%). This result is the most reliable because it comes from the majority of the unsafe behavior data. The rest of the data in this category (33.63 %) are linked to other categories except for category H. There are some rules as the result from the Apriori algorithm. Rule 1 shows that 44.11% of Passage and lighting category is linked to construction equipment inspections. Rule 2 also shows that 29.41% of High-pressure gas is linked to construction equipment inspections. Rule 3 indicates that 100% of Fire prevention category is linked to Fire Protection unsafe behavior, while in rule 4 it can be seen that 71.42% of Lifting and hanging work is linked to Lifting Operation. Rule 5 is also linked to Lifting Operation unsafe behavior which has 23.8% as confidence value of Machinery category.

For the unsafe behavior that does not belong to regulation and laws VII, the result indicates that only regulation and laws II and VIII have significant support and confidence values. Rule 1 shows that 100% of Self-inspection affects the construction equipment inspections, which refers to any error in the Self-inspection would be linked to Construction equipment inspections. This rule is also linked to rule 2, that 100% of Occupational Safety and Health Management Measures violation is about Construction equipment inspections. In Rule 3, it can be seen that 100% of Personal misconduct is about alcoholic beverages. The Personal misconduct is also 100% linked to Standards for Labor Protection Measures for Elevated Work violation (Rule 4). This can be concluded that any unsafe behavior in a construction project would attribute to alcoholic beverages and the violation of the regulation and laws II; Standards for Labor Protection Measures for Elevated Work. Rule 5 and Rule 6 have relatively low number of support (below 5%), but it has the perfect score of confidence (100%). Rule 5 indicates that an excavation operations category originates from excavation operations unsafe behavior; while Rule 6 indicates that a Lifting Operation unsafe behavior would affect Lifting and hanging work category. As stated above, based on statistical and Apriori algorithm output, significant unsafe behavior pat-terns are as follows:

  1. Regulations for the Occupational Safety and Health Equipment and Measures ® Fall prevention ® Fall Protection and Structure Strengthening
  2. Fire prevention ® Fire Protection.
  3. Confined space operation ® Safety management and equipment.
  4. Safety management and equipment, and Construction equipment inspections ®High-pressure gas.
  5. Self-inspection ® Construction equipment inspections.
  6. Occupational Safety and Health Management Measures ® Construction equipment inspections.
  7. Personal misconduct ® Alcoholic beverages.
  8. Personal misconduct ® Standards for Labor Protection Measures for Elevated Work.
  9. Excavation operation ® Excavation operation.
  10. Lifting operation ®Lifting and hanging work
  11. Safety management and equipment ®
  12. Safety management and equipment ® Educational training.
  13. Occupational Safety and Health Act ® Safety management and equipment.

Please see the highlighted parts on Pages 7 and 8.

  1. The conclusion fails to tie together the various elements of the paper. There is the need to discuss the theoretical and practical implications of the paper. Also, key limitations must be stated and recommendations for further studies must be made.

Responses: We have re-written the conclusion section by “The case study toward construction sites regarding accident prevention in Taiwan aims to investigate the pattern of unsafe behaviors in order to uncover the relationship between unsafe behaviors and the violation of regulation and laws. A total of 2,207 observations of unsafe behavior were recorded based on convenient sampling with 95% confidence and 5% limits of errors in 50-50 category. There were 1,775 unsafe behaviors (or 80.43% out of 2,207 reports) categorized into “Regulations for the Occupational Safety and Health Equipment and Measures”. Furthermore, in this category there were 1,178 unsafe behaviors (66.37% of regulation and laws VII violation) that were linked to category H with the most cases of Fall Protection and Structure Strengthening (94.48%). These are the most reliable because it comes from the majority of the unsafe behavior data. The Apriori yields 13 association rules. For example, Rule 1 shows that 44.11% of Passage and lighting category is linked to Construction equipment inspections. Rule 2 also shows that 29.41% of High-pressure gas is linked to Construction equipment inspections. Rule 3 indicates that 100% of Fire prevention category is linked to Fire Protection unsafe behavior, while in rule 4 it can be seen that 71.42% of Lifting and hanging work is linked to Lifting Operation. Rule 5 is also linked to Lifting Operation unsafe behavior which has 23.8% as confidence value of Machinery category. The contribution lies in the association rules that guide management practice for construction job sites.

The current domestic occupational safety laws and regulations were designed to protect the lives of workers; however, the regulations themselves affect construction efficiency and increase construction costs, resulting that domestic workers in construction sites may be unable to properly follow SOPs. Findings from this study provide priority suggest that those 13 rules should serve as guideline to set up priority in order to simply the currents SOPs. It especially takes place when safety budget for construction projects is limited, implying that safety management has priority in practice. Future work can focus on how to classify the safety inspections in order to practice more efficiently. Creating a system that conducts automation detection applied to workers’ behaviors in order to send out warnings to those behaving unsafe postures or patterns is also feasible and contributable to the industries.”. Please see the highlighted parts on Page 8.

  1. The paper with the details below could help improve the literature review and other sections as well:

Adinyira, E., Manu, P., Agyekum, K., Mahamadu, A. M., & Olomolaiye, P. (2020). Violent Behaviour on Construction Sites: Structural Equation Modelling of its Impact on Unsafe Behaviour Using Partial Least Squares. Engineering Construction and Architectural Management. https://doi.org/10.1108/ECAM-09-2019-0489

Responses: We have added the reference in the literature review. Please see the highlighted reference of [22] on Page 10.

Reviewer 3 Report

1. Are you proposing any new algorithm?
2. Give the mathematical formulas for proposed algorithm.
3. You can compare the results with some other latest methodologies and try to prove your algorithms strength.
4. What are the performance metrics can be considered for the comparison part?
5. What are the steps involved in data pre-processing? Justify with proper statistical formulas.

Author Response

Reviewer: 3

  1. Are you proposing any new algorithm?

Responses: No, Apriori is not a new algorithm. We have reorganized the sections to: 1. Introduction, 2. Literature Review (2.1. Safety in construction, 2.2. Data mining in construction health and safety research, 2.3. Association rule mining in construction health and safety research), 3. Site Investigation and Observation Sampling, 4. Association Rule Mining, 5. Research Findings and Discussions, 6. Conclusions”. Please see the highlighted parts on Pages 3-8.

  1. Give the mathematical formulas for proposed algorithm.

Responses: Since the Apriori method was introduced years ago, this study presents it in the Sections 2.2 and 2.3 by “2.2. Data mining in construction health and safety research

In the construction business, the use of data mining approaches is becoming more popular. It has attracted the attentions of both practitioners and academics around the world [21,24]. It also can be learned from vast amounts of objective, empirical data equating to millions of labor hours [23] to optimize safety management methods [25]. Poh et al. applied a machine learning approach to create a model that can produce a safety leading indicator as well as be used to forecast building site safety concerns. The industry-recognized Cross Industry Process Model for Data Mining (CRISP-DM) framework was used to systematically apply Machine Learning (ML) procedures and techniques to a seven-year dataset of a big and reputable construction company. A total of thirty-three input variables (also known as features or in-dependent variables) were identified as part of the research. Thirteen relevant input variables were identified after using the Boruta feature selection technique and a decision tree. Six of the thirteen input variables are project-related (Project Type, Project Ownership, Contract Sum, Completion Percentage, Magnitude of Delay, and Project Manpower) and seven are safety-related (Crane/lifting Operations, Scaffold, Mechanical-Elevated Working Platform, Falling Hazards/Openings, Environmental Management, Good Practices, and Weighted Safety Inspection Score). This emphasizes the significance of good project management in terms of construction safety [21]. Sakhakarmi and his team investigated the effect of feature vectors in classifying likely structural failure scenarios of a complicated scaffold structure using machine learning. Over 23,000 datasets, 23 forms of local and global failures were studied, with 20 original strain characteristics per dataset and 210 enhanced strain features per dataset. Based on the simulation findings, 20 strain features were found to be insufficient to adequately reflect the failure instances caused by complicated loading combinations given to each of the datasets. The greatest accuracy with the 20 strain feature case was 85.48 percent, whereas it was 96 percent with the 210 strain feature instance. The influence of enhanced strain feature vectors for exactly the same loading scenarios was seen when the prediction accuracy was compared between the two cases [26]. Tixier et al. adopted two cutting-edge ML algorithms: Random Forest (RF) and Stochastic Gradient Tree Boosting (SGTB). The generated models predict three out of four safety outcomes with high skill, namely injury type, energy type, and body part, using binary fundamental construction qualities as input. This model not only outperforms previous research models in terms of competence but also in terms of the range of outcomes anticipated. It's also worth noting that the SGTB models consistently outperformed their RF counterparts in terms of predictive ability [23].

2.3. Association rule mining in construction health and safety research

Association Rule Mining (ARM) is a technique for extracting common patterns, correlations, relationships, and causal structures from datasets stored in a variety of databases and repositories. Ithas been used to characterize and analyze unknown relationships in data of safety concerns assessment, which is for identifying frequent item sets between uncertain parameters and generating strong association rules from huge datasets, especially auxiliary data from engineering processes. This simple and practical data mining technique can uncover common item sets and association rules across many atypical events that have been tracked [27-30]. The Apriori method was used to analyze the injury data in order to find statistical correlations between injuries (IBIP). The outcome variable for each IBIP was then employed in logistic regression modeling to find connections between specific road user groups and IBIPs. A total of 48,544 people were studied, with 36,480 (75.1%) having only one injury type and 12,064 (24.9%) having multiple injuries. In the multiple injury sample, data mining revealed 77 IBIPs, 16 of which were linked to only one type of road user. IBIPs and their relationship to the type of road user are one step toward establishing a tool to better comprehend and quantify injury severity and, as a result, improve the evidence base supporting the prioritization of road safety countermeasures [31]. Sivasankaran et al. employed a data mining approach to evaluate vehicle-pedestrian collisions in order to extract information that may be used to improve pedestrian safety. The non-parametric method of association rule mining has the distinct advantage of not limiting the distribution assumption of variables and associations, whereas parametric modeling requires that explanatory variables be independent. Additional association rules uncover significant rules with specific patterns that are not possible in traditional statistical models [32]. Martínez-Rojas et al. examined workplace accidents that occurred on Spanish construction sites while taking into account the workers' nationality. The usage of association rules is suggested due to the vast number of incidents and attributes linked with them. Overall, the results show comparable behavior, however there are some notable variances in the workers' occupations. Furthermore, the findings are consistent with earlier research conducted in other nations. The results of these investigations would be used to develop initiatives to create safer working environments [33]. However, occupational safety accidents still frequently occur on construction sites, both in public and private projects. As a result, construction site workers are reluctant to comply with the laws and regulations, and often make mistakes during construction that do not comply with the laws and regulations, which greatly increases the occurrence rate of accidents [34].”. Please see Sections 2.2 and 2.3 on Pages 2 and 3.

  1. You can compare the results with some other latest methodologies and try to prove your algorithms strength.

Responses: We have introduced the Apriori practicability on construction health and safety research on Section 2.3 by “Association Rule Mining (ARM) is a technique for extracting common patterns, correlations, relationships, and causal structures from datasets stored in a variety of data-bases and repositories. Ithas been used to characterize and analyze unknown relation-ships in data of safety concerns assessment, which is for identifying frequent item sets between uncertain parameters and generating strong association rules from huge datasets, especially auxiliary data from engineering processes. This simple and practical data mining technique can uncover common item sets and association rules across many atypical events that have been tracked [27-30]. The Apriori method was used to analyze the injury data in order to find statistical correlations between injuries (IBIP). The outcome variable for each IBIP was then employed in logistic regression modeling to find connections between specific road user groups and IBIPs. A total of 48,544 people were studied, with 36,480 (75.1%) having only one injury type and 12,064 (24.9%) having multiple injuries. In the multiple injury sample, data mining revealed 77 IBIPs, 16 of which were linked to only one type of road user. IBIPs and their relationship to the type of road user are one step toward establishing a tool to better comprehend and quantify injury severity and, as a result, improve the evidence base supporting the prioritization of road safety counter-measures [31]. Sivasankaran et al. employed a data mining approach to evaluate vehicle-pedestrian collisions in order to extract information that may be used to improve pedestrian safety. The non-parametric method of association rule mining has the distinct advantage of not limiting the distribution assumption of variables and associations, whereas parametric modeling requires that explanatory variables be independent. Additional association rules uncover significant rules with specific patterns that are not possible in traditional statistical models [32]. Martínez-Rojas et al. examined workplace accidents that occurred on Spanish construction sites while taking into account the workers' nationality. The usage of association rules is suggested due to the vast number of incidents and attributes linked with them. Overall, the results show comparable behavior, however there are some notable variances in the workers' occupations. Furthermore, the findings are consistent with earlier research conducted in other nations. The results of these investigations would be used to develop initiatives to create safer working environments [33]. However, occupational safety accidents still frequently occur on construction sites, both in public and private projects. As a result, construction site workers are reluctant to comply with the laws and regulations, and often make mistakes during construction that do not comply with the laws and regulations, which greatly increases the occurrence rate of accidents [34].” Please see Section 2.3 on Page 3.

  1. What are the performance metrics can be considered for the comparison part?

Responses: The Apriori method is a well-applied method in data mining that assists practitioners in finding association rules to deal with complicate problems. We have added more discussions for metrics in Section 5 by “The occurrence of unsafe behavior is often caused by a number of factors, including the manufacturer's own costs, the convenience of the manufacturer, the difficulty of construction due to regulations, or the convenience of the laborers themselves, all of which can cause errors. The statistical categories are shown in Figure 3. Figure 3 shows the categories of each unsafe behavior that were sorted into construction method, self-inspection, and (one more example), etc. The statistics show that the first two most frequent unsafe behavior samples in construction sites are fall prevention (55%) and Scaffolding (23%). Figure 3 also indicates that 80% of unsafe behaviors are due to the violation of regulation & laws VII, which is Regulation for the Occupational safety and Health Equipment and Measures. More specifically, reinforcing steel and formwork workers are the most common types of unsafe behavior violating this regulation. Based on this finding, the unsafe behaviors are structured based on regulation and laws as shown in Figure 2 where the Apriori setting for confidence is greater than 90%.

There were 1,775 unsafe behaviors (or 80.43% out of 2,207 reports) categorized into “Regulations for the Occupational Safety and Health Equipment and Measures”. Furthermore, in this category there were 1,178 unsafe behaviors (66.37% of regulation and laws VII violation) that were linked to category H with the most cases of Fall Protection and Structure Strengthening (94.48%). This result is the most reliable because it comes from the majority of the unsafe behavior data. The rest of the data in this category (33.63 %) are linked to other categories except for category H. There are some rules as the result from the Apriori algorithm. Rule 1 shows that 44.11% of Passage and lighting category is linked to construction equipment inspections. Rule 2 also shows that 29.41% of High-pressure gas is linked to construction equipment inspections. Rule 3 indicates that 100% of Fire prevention category is linked to Fire Protection unsafe behavior, while in rule 4 it can be seen that 71.42% of Lifting and hanging work is linked to Lifting Operation. Rule 5 is also linked to Lifting Operation unsafe behavior which has 23.8% as confidence value of Machinery category.

For the unsafe behavior that does not belong to regulation and laws VII, the result indicates that only regulation and laws II and VIII have significant support and confidence values. Rule 1 shows that 100% of Self-inspection affects the construction equipment inspections, which refers to any error in the Self-inspection would be linked to Construction equipment inspections. This rule is also linked to rule 2, that 100% of Occupational Safety and Health Management Measures violation is about Construction equipment inspections. In Rule 3, it can be seen that 100% of Personal misconduct is about alcoholic beverages. The Personal misconduct is also 100% linked to Standards for Labor Protection Measures for Elevated Work violation (Rule 4). This can be concluded that any unsafe behavior in a construction project would attribute to alcoholic beverages and the violation of the regulation and laws II; Standards for Labor Protection Measures for Elevated Work. Rule 5 and Rule 6 have relatively low number of support (below 5%), but it has the perfect score of confidence (100%). Rule 5 indicates that an excavation operations category originates from excavation operations unsafe behavior; while Rule 6 indicates that a Lifting Operation unsafe behavior would affect Lifting and hanging work category. As stated above, based on statistical and Apriori algorithm output, significant unsafe behavior pat-terns are as follows:

  1. Regulations for the Occupational Safety and Health Equipment and Measures ® Fall prevention ® Fall Protection and Structure Strengthening
  2. Fire prevention ® Fire Protection.
  3. Confined space operation ® Safety management and equipment.
  4. Safety management and equipment, and Construction equipment inspections ®High-pressure gas.
  5. Self-inspection ® Construction equipment inspections.
  6. Occupational Safety and Health Management Measures ® Construction equipment inspections.
  7. Personal misconduct ® Alcoholic beverages.
  8. Personal misconduct ® Standards for Labor Protection Measures for Elevated Work.
  9. Excavation operation ® Excavation operation.
  10. Lifting operation ®Lifting and hanging work
  11. Safety management and equipment ®
  12. Safety management and equipment ® Educational training.
  13. Occupational Safety and Health Act ® Safety management and equipment.

Please see the highlighted parts on Pages 7 and 8.

  1. What are the steps involved in data pre-processing? Justify with proper statistical formulas.

Responses: We have added descriptions for site investigation and observation sampling by “The study conducted the site investigation and observation sampling to investigate the frequency and types of unsafe behaviors that occur at the construction sites, and to obtain the unsafe behavior patterns that occur at the sites through on-site surveys. The target for this study was, based on the concept of the convenient sampling [35], to select the largest construction area in Taoyuan City, Taiwan during 2019-2021. There are dozens of construction sites spreading out for over 50,000 square meters of area in Taoyuan Technological and Environmental Industrial Park, Taoyuan City, Taiwan. The total construction cost was around 10 billion New Taiwan Dollars (NTD). During 2019-2021, the sites were visited and observed to record unsafe behaviors for on-site workers, who conducted their work at job sites illustrated in Figure 1. This study adopted the participatory observation method as the main collection method where the observers served as occupational safety inspectors conducted on-site investigation and data collection at the sites including a major incinerator construction project. During the observation period of over 2 years, the supervisors and occupational safety and health personnel in charge of each area of the site carried out daily inspections to investigate and compiled records of errors and violations in accordance with the laws and regulations. They are Occupational Safety and Health Law, the Occupational Safety and Health Facilities Regulations, the Occupational Safety and Health Facilities Standards, and other related regulations promulgated by the Minis-try of Labor Affairs.

The observations were conducted in a floor-by-floor walk-through manner, with outdoor excavation areas and flat areas and indoor areas such as basements and flat areas, and floor-by-floor walk-throughs. In accordance with the regulations and inspection procedures related to occupational safety and health at construction sites, safety and health personnel were required to conduct toolbox safety education, hazard notification signatures, and toolbox meetings with on-site workers who work in the investigated job sites not including supervision manpower, and to take photos of occupational safety and health errors seen during construction. Figure 1 demonstrates that workers conducted unsafety behaviors including no protective actions in aloft activities and grouting. A total of 2,207 observations of unsafe behavior were recorded based on convenient sampling with 95% confidence and 5% limits of errors in 50-50 category [36, 37]. The study then compared and analyzed the contents of the current occupational safety and health laws and regulations with the common unsafe behaviors, explored the discordance between the unsafe behavior and the occupational safety laws and regulations.” Please see the highlighted parts on Pages 3 and 4.

Round 2

Reviewer 1 Report

Most comments and queries have been well addressed, so I think this version has meet the requirement to be published. 

Reviewer 2 Report

I wish to thank the authors for taking their time to improve the paper. Congratulations.

Reviewer 3 Report

Authors have submitted their response nicely. This paper may be accepted.